# Antagonistic Bacteria *Bacillus velezensis* VB7 Possess Nematicidal Action and Induce an Immune Response to Suppress the Infection of Root-Knot Nematode (RKN) in Tomato

**DOI:** 10.3390/genes14071335

**Published:** 2023-06-25

**Authors:** Vinothini Kamalanathan, Nakkeeran Sevugapperumal, Saranya Nallusamy

**Affiliations:** 1Department of Plant Pathology, Centre for Plant Protection Studies, Tamil Nadu Agricultural University, Coimbatore 641 003, Tamil Nadu, India; vinokjc19@gmail.com; 2Department of Plant Molecular Biology and Bioinformatics, Centre for Plant Molecular, Biology and Biotechnology, Tamil Nadu Agricultural University, Coimbatore 641 003, Tamil Nadu, India; saranya.n@tnau.ac.in

**Keywords:** tomato, *M. incognita*, *B. velezensis* VB7, qRT PCR

## Abstract

*Meloidogyne incognita*, the root-knot nematode (RKN), a devastating plant parasitic nematode, causes considerable damage to agricultural crops worldwide. As a sedentary root parasite, it alters the root’s physiology and influences the host’s phytohormonal signaling to evade defense. The sustainable management of RKN remains a challenging task. Hence, we made an attempt to investigate the nematicide activity of *Bacillus velezensis* VB7 to trigger the innate immune response against the infection of RKN. In vitro assay, *B. velezensis* VB7 inhibited the hatchability of root-knot nematode eggs and juvenile mortality of *M. incognita* by 87.95% and 96.66%, respectively at 96 hrs. The application of *B. velezensis* VB7 challenged against RKN induced MAMP-triggered immunity via the expression of transcription factors/defense genes by several folds pertaining to WRKY, LOX, PAL, MYB, and PR in comparison to those RKN-inoculated and healthy control through RT-PCR. Additionally, Cytoscape analysis of defense genes indicated the coordinated expression of various other genes linked to immune response. Thus, the current study clearly demonstrated the effectiveness of *B. velezensis* VB7 as a potential nematicide and inducer of immune responses against RKN infestation in tomato.

## 1. Introduction

Globally, root-knot nematodes (*Meloidogyne* spp.) are regarded as the most significant of the plant-parasitic nematodes that infect horticultural crops. *Meloidogyne* spp. have been observed on different crop plants since the 19th century [1]. Infestation by RKN causes severe damage and yield reduction, accounting for a 15–35% loss in crop plants [2,3], and is responsible for substantial economic losses throughout the world [4]. Despite being a sedentary obligate parasite, RKNs are difficult to eradicate as they have a wide host range. The association of *Meloidogyne* spp. with host plants leads to the formation of galls in infected regions, wilting, and stunted growth [5,6,7]. As an obligate root feeder, it spends most of its life within the host roots, which affects root physiology and influences the host’s phytohormonal signaling to evade defenses and generate a nutritional sink. Many strategies, including chemical, cultural, and biological methods, have been employed to manage plant parasitic nematodes. However, the use of chemical nematicides alone for nematode management has a severe environmental impact and becomes less effective over time, prompting the development of alternative sustainable strategies. Consequently, the use of bioagents has been considered as an alternative approach for the management of plant parasitic nematodes [8,9]. Among the biological agents, microorganisms belonging to the genera *Bacillus*, *Paenibacillus*, *Pseudomonas*, *Streptomyces*, *Alcaligenes*, *Agrobacterium*, *Serratia*, *Clostridium*, and *Desulfovibrio* have been known to possess nematicidal properties [10]. Owing to the complex endophytic bacteria–plant networks and co-occurrence relationships in roots, and due to the ability to enhance the immune response against nematode infection, bacterial endophytes have been explored to curb root-knot nematode infection [11]. They promote plant growth and counteract the plant’s parasitic nematodes through the production of phytohormones, antibiotics, endospores, hydrolytic enzymes, and volatile organic compounds (VOCs) and enhance the uptake of nutrients from the soil [12]. Several *Bacillus* species such as *B. velezensis* [13,14], *B. pumilus* [15], *B. megaterium* [16], *B. subtilis* [17], *B. firmus* [18], *B. thuringiensis* [19], and *B. amyloliquefaciens* [20] have been reported to be effective biocontrol agents and activate Systemic Acquired Resistance (SAR) and Induced Systemic Resistance (ISR) against root-knot nematodes. Extensive research into their molecular biology has proven that beneficial bioagents can stimulate plant defense mechanisms by releasing elicitors that activate the plant transcription factors and defense genes involved in resistance to biotic and abiotic stresses [21]. In SAR, the hormone SA utilizes the redox-regulated protein Non-Expressor of Pathogenesis-Related Genes 1 (NPR1) to stimulate PR (pathogenesis-related) genes, a major group of genes associated with plant defense [22], whereas jasmonic acid (JA) and ethylene (ET) signaling play a significant role in regulating rhizobacterial-mediated ISR [23]. *M. incognita* triggered both local and systemic defense responses in tomato plants by the applications of bioagents [24]. During the interaction, several biomolecules with antifungal and nematicidal properties were induced in the host plants. As a consequence, we attempted to investigate the differential expression of five crucial and key defense genes in tomato plants during nematode interaction by the application of *B. velezensis* VB7 through quantitative real-time polymerase chain reaction (qRT-PCR) in order to better understand the resistance mechanisms against RKN. The complex signaling network initiated upon pathogen attack enables the triggering of appropriate cellular mechanisms to defend the pathogen invasion by transcriptional mechanisms that are directly linked with physical interactions between proteins (interactome) [25,26]. The protein–protein interaction (PPI) would be essential to actively regulate or modify the resistance pathways as well as to select combinations of resistance genes that enhance their durability. In addition, it also mediates the effector recognition, protein phosphorylation, and transcriptional co-factor activation process. In the current study, the application of *B. velezensis* VB7 during RKN infection was assessed for its potent nematicide activity and its potential to reprogram the immune response to prevent RKN infection in tomato plants.

## 2. Materials and Methods

### 2.1. Isolations and Morphological Identifications of Root-Knot Nematode from Root Galls of Tomato

Tomato plants infested with nematodes were collected from four locations in the Coimbatore district of Tamil Nadu, India: Thondamuthur (11°00′35′′ N 76°49′41′′ E; latitude 10.9905, longitude 10.9905), Kuppanur (10.94′78.35′′ N, 76°86′27.27′′ E), Thaliur (11°19′16.9′′ N, 77°0′18.8′′ E; latitude 11.3183, longitude 77.0066) and Mathampatti (latitude 10.9816015, longitude 76.8513025). Posterior cuticular patterns (PCP) of the isolated RKNs were observed for the identification of nematodes [27,28]. The single matured female was carefully dissected out from the roots using a fine needle, forceps, and scalpel. The posterior-most region of nematodes showing the vulval region was dissected and trimmed carefully without disturbing the perennial pattern. The inner body contents were removed and mounted on a glass slide containing dehydrated glycerol and observed under a microscope. Isolated nematodes from the field were maintained as a pure culture under greenhouse conditions at the Department. of Plant Pathology, Tamil Nadu Agricultural University for further studies.

### 2.2. Molecular Characterization of Root-Knot Nematode

Molecular characterization was carried out to confirm the nematode species. Methodology reported by [29] was followed for the isolation of DNA from individual females and juveniles (J2) using a worm lysis buffer. The polymerase chain reactions were carried out using the primers of TW 81 (F)-GTTTCCGTAGGTGAACCTGC AB 28(R)-ATATGCTTAAGTTCAGCGGGT. The PCR mixtures for 40 µL, containing 2 µL of template DNA, 20 µL of 2X Taq PCR Master Mix (Takara, Shiga, Japan), 2 µL of 10 µM each primer, and 16 µL of ddH2O. The PCR cycle included the following steps: initial denaturation at 94 °C for 5 min, followed by 35 cycles at 94 °C for 30 s, annealing temperature of 56 °C for 45 s and extension with 72 °C for 1 min, with a final extension step for 10 min at 72 °C. A measure of 10 μL of PCR product was loaded onto a 1% stained agarose gel in TAE buffer and electrophoresed at 75 V for 45 min at 400 Ampere. To quantify the size of the amplified genomic products, a 100 bp DNA ladder was utilized, and the gel photographs of the PCR results were documented. The amplified genomic product was sequenced by Eurofins Genomics Biotech Pvt. Ltd., Bangalore, India. Gene homology searches were performed using NCBI BLAST. Sequences were compared with different *M. incognita* isolates retrieved from the GenBank database. Newly obtained sequences were submitted to the GenBank database (New York, NY, USA), and accession numbers were obtained. Phylogenetic analysis was performed with MEGA7 software [30].

### 2.3. Preparation of Nematode Inoculum

Eggs collected from severely infected galled tomato roots were used as nematode inoculum. To separate the eggs from the gelatinous matrix, the roots were chopped into pieces of 1–2 cm, placed in a 500 mL plastic container, filled with 1.5% chlorine solution, and shaken violently for 3 min. The suspension was then rinsed many times with running water through a 250 mm sieve, the eggs being collected on a sieve with a mesh size of 20 mm. After 4 days of incubation at 25 ± 2 °C, the hatched second-stage juveniles (J2) were collected from the egg suspension using a modified Baermann dish. The hatched second-stage juveniles (J2) were collected and utilized for the experiments.

### 2.4. Testing the Nematicidal Activity of Bacterial Endophytes against RKN

The six bacterial endophytes such as *B. velezensis* VB7 (MW301630), *B. velezensis* (MW301615), *B. subtilis* (MW301629) *B. haynesii* (MW301614), *B. licheniformis* (MW331690), *B. subtilis* (MW331691) were obtained from the Department of Plant Pathology, Tamil Nadu Agricultural University (TNAU), Coimbatore. The effect of the bacterial strains with nematicide activity on the second-stage juveniles (J2) of the *M. incognita* was evaluated by their mortality rate and the hatching ability of the egg mass. The bacterial culture was inoculated into LB broth and maintained in an orbital shaker at 150 rpm at room temperature (28 ± 2 °C) for 48 h to ensure the uniform bacterial growth. Supernatants were collected by centrifugation of bacterial suspension at 10,000 rpm for 10 min, which contained no viable bacteria. The supernatants were diluted at a ratio of 1:1, 1:5, 1:10 and 1:15, using sterile water. A measure of 3 mL of diluted cell-free bacterial suspension was transferred into a 6 cm Petri dish and inoculated with a 2-egg mass for each replication with periodical aeration. The Petri dishes were incubated at 28 ± 2 °C at room temperature, and the percentage of hatching were observed at 24 h, 48 h, 72 h, 96 h after inoculation. Similarly, for mortality assay, hatched second-stage juveniles (J2) were adjusted to the concentration of 50 juveniles mL^−1^. A measure of 2 ml of nematode suspension (100 juveniles) was inoculated into 6 cm Petri plates containing a 3 mL cell-free bacterial suspension. Three replications were maintained and incubated at room temperature (28 ± 1 °C). After 24, 48, 72 and 96 h, the number of surviving and dead individuals were recorded using a 1 ml Hawksley counting slide. Percentage mortality was calculated by following the formula Mortality (%) = [C_1_ − C_2_/C_1_] × 100, where C_1_ is the number of live juveniles released, and C_2_ is the number of alive juveniles counted. Analysis of variance was used to conduct a statistical analysis of the data collected from in vitro experiments. For analysis of variance (ANOVA), DMRT was performed using SPSS 20.0 (IBM, SPSS statistics 20) with a significance threshold of 0.05.

### 2.5. Preparation and Applications of Liquid Formulations of B. velezensis VB7 and Inoculation of M. incognita

The bacterial inoculum used for this experiment was prepared as per the standard protocol [31]. The single colony of bacteria was inoculated into LB broth, which was then maintained in an orbital shaker at 150 rpm at room temperature (28 ± 2 °C) for 48 h to ensure uniform bacterial growth. The culture broth was mixed with 1% glycerol (10 mL), 1% tween 20 (10 mL), and 1% poly vinylpyrrolidone (10 g). The mixture was incubated in an orbital shaker at 200 rpm for 5 min to ensure uniform mixing. The bacterial suspension of the liquid formulation was adjusted to 5 × 10^8^ CFU mL^−1^ [32]. The pots were filled with a sterilized potting mixture containing red soil, sand, and cow dung manure at 1:1:1 *w*/*w*/*w*. Later, 20-day-old tomato seedlings were transplanted into a pot. After 5 days of transplanting, the soil was drenched with a liquid formulation of *B. velezensis* VB7 at 1% suspensions (5 × 10^8^ cfu/mL). After transplanting the tomato seedlings to pots, a hole was made by inserting 4 sterile glass rods around the plants. The rods were removed after 2 days, and a juvenile suspension of one juvenile/gram of soil was injected into the holes and covered with autoclaved soil.

### 2.6. RNA Extraction and Quantitative Real-Time PCR

A real-time polymerase chain reaction (RT-PCR) was performed with the RNA samples extracted from the tomato plants at 0, 3, 5, and 7 days after treatment with *B. velezensis* VB7 challenged with RKN to confirm the differential expression of selected defense genes/transcription factors in different treatments. For every treatment, three replications were maintained; for each replication, five plants per pot were maintained. Twenty-day-old tomato hybrid (Shivam) seedlings were transplanted at five seedlings per pot in the potting mixture. Five days after planting, seedlings were drenched with 1% *B. velezensis* VB7. Total RNA was isolated from the rhizosphere samples pertaining to eight different treatments, including T_1_-Healthy control, T_2_-Inoculated control (RKN), T_3_-*B. velezensis* VB7, and T_4_-*B. velezensis* VB7 + RKN at different intervals on 0, 3, 5, and 7 DAI, using Trizol reagent (Sigma Aldrich) [33]. Furthermore, the quality of RNA was analyzed using a nanodrop at an absorbance ratio of A260/A280. A Thermo Fischer Scientific-Revert Aid First Strand cDNA Synthesis Kit (Catalogue number K1622) was used to synthesize cDNA from RNA. A good quality nucleic acid ratio of 1.8 + 2.0 was used for qRT-PCR analysis. Quantitative real-time polymerase chain reaction was performed using cDNA, diluted to 10-fold. The BIORAD CFX manager system was used to perform the experiment. The qRT-PCR reaction was carried out for 20 µL volume containing 3 µL of cDNA template, 10 µL of SYBR Green master mix (KAPA SYBR @ FAST for Light Cycler 480, Cat-KK4610), 0.8 µL of forward primer, and 0.8 µL of reverse primer, at a concentration of 10 mM. The protocol for the PCR reaction involved initial denaturation at 95 °C for 10 min, denaturation at 95 °C for 30 s with amplification of 40 cycles, 58 °C for 30 s (Annealing), and 72 °C for 30 s (extension), respectively. The standard melting temperature was followed for analysis. The defense genes, including Pathogenesis Related Protein-1 (PR1), Lipoxygenase (LOX), Phenylalanine Ammonia Lyase (PAL), and the TFs of WRKY33 and MYB were selected for the analysis of MAMP-triggered immunity by drenching the soil with *B. velezensis* VB7 @ 10^8^ cfu/mL against RKN infection in tomato. Three biological replicates and two technical replicates were maintained for every individual gene for its expression studies. Statistical analysis was carried out to find the fold changes in gene expression using the formula ΔΔCt = ΔCt sample − ΔCt reference. The relative fold changes in the transcript level were represented graphically by converting the ΔΔCT value to 2^−ΔΔCT^ [34]. The software TIBCO Spotfire Analyst 7.11.1 was used for the statistical analysis of relative fold change.

### 2.7. Protein-Protein Interaction

Defense-related proteins and transcript genes in tomato plants play a significant role in protecting tomato plants against RKN infection. Protein–protein interactions were examined using the STRING (Search Tool for the Retrieval of Interacting Genes/Proteins database version 10.0) database [35]. The co-expression of proteins, a fusion of genes, and data mining are implemented in the STRING database to determine the interacting functional proteins, the number of protein domains, as well as their associating partners in a particular domain for understanding the functional relevance. The protein–protein interaction network was analyzed for defense-regulating genes, including transcription factors and their interacting proteins, including PR1, WRKY33, MYB, LOX and PAL through Cytoscape. Cytoscape (3.9 version) software was used for visualizing molecular interaction networks and integrating these interactions with gene expression profiles for analyzing the data. Functional enrichment was performed using the STRING enrichment map tool for the chosen ten proteins and their associated partners (Saravanan et al. 2022).

### 2.8. Statistical Analysis

All the experiments were analyzed independently. The treatment means were compared by Duncan’s Multiple Range Test (DMRT) (Gomez and Gomez 1984). The package used for analysis was SPSS version 16.0., developed by IBM Corporation, with a critical difference at *p* = 0.05 and interpreted.

## 3. Results

### 3.1. Morphological and Molecular Confirmation of RKN

The matured female *M. incognita* was excised for the preparation of the PCP. The perineal pattern of all four *M. incognita* females appeared with typical characteristics, including round to oval-shaped, high dorsal arch with substantial wavy striae and no lateral lines under a microscope (Figure 1A,B). The isolates of the root-knot nematode *M. incognita* were characterized through PCR using the 18S rRNA gene. The amplicon size of approximately 600 bp was amplified, and the PCR product was resolved on 1.2% agarose gel (Figure 2A). The amplified and purified genomic product was sequenced by Eurofins Genomics Biotech Pvt. Ltd., Bangalore, India. The sequence analysis of four RKN isolates using NCBI BLAST revealed a nucleotide sequence homology of 50–100% with their existing isolate. Based on the sequences, the nematode was confirmed as *M. incognita,* and the same was submitted to NCBI GenBank with the accession numbers MW662288, MW662261, OQ129945, and OQ121829. The phylogenetic analysis of RKN isolates confirmed the presence of three different clusters. The Neighbour-Joining tree of 18S rRNA sequences of *M. incognita* isolates was constructed with bootstrap values of more than 500 and indicated that the nodes as a percentage vary with each cluster (Figure 2B).

### 3.2. Efficacy of Culture Filtrate of Bacterial Endophytes against Egg Hatching and Juveniles’ Mortality of Root Knot Nematode (RKN) M. incognita In Vitro

The six bacterial endophytes such as *B. amyloliquefaciens* (MW301630), *B. velezensis* (MW301615), *B. subtilis* (MW301629), *B. haynesii* (MW301614), *B. licheniformis* (MW331 690), *B. subtilis* (MW331691) were screened against egg hatching and juvenile mortality in vitro. The cultural filtrates of bacterial endophytes at different concentrations, *viz*., 1:1, 1:5, 1:10, and 1:15 dilutions, were screened for their efficacy on egg hatching and juveniles’ mortality. Among them, a 1:1 dilution concentration of *B. velezensis* VB7 effectively inhibited 97.50% of the egg hatching and 87.65% of the juvenile mortality of *M. incognita* after 96 h exposure, followed by *B. haynesii*, which accounted for an 81.13% inhibition of egg hatching and 75.33% of J2 mortality (Figure 3A,B). The other isolates inhibited RKN in the range of 75% to 56% and 69% to 54% pertaining to egg hatching inhibition and juveniles’ mortality, respectively. The lowest inhibitions of 65.27% for egg hatching and 54.66% for juveniles mortality were recorded from *B. subtilis* (MW331691) in comparison with control (Appendix A).

### 3.3. Induction of MAMP Triggered Immunity by B. velezensis for Expression of Defense Genes in Tomato against RKN

The current investigation was conducted to determine whether the expression levels of PR1, WRKY33, MYB, LOX, and PAL were influenced by *B. velezensis* VB7 with or without RKN challenge inoculation in tomato plants using different treatments.

### 3.4. WRKY 33

The transcript level of transcription factor WRKY 33 increased in all the treatments at 3 DAI and was maintained up to 7 DAI. Even at 0 DAI, in all the treatments it was upregulated, except in the RKN-inoculated plants. The transcript level increased 2.51-fold in *B. velezensis* VB7 challenged with RKN, followed by 1.89-fold increases in *B. velezensis* VB7 without RKN when compared to both inoculated control (0.46-fold) and healthy control (0.3-fold) at 7 DAI. A comparison with the other treatments indicated that *B. velezensis* VB7-inoculated plants with or without RKN showed a greater level of expression in all the intervals than RKN-inoculated control as well as healthy control. However, the expression level of WRKY 33 was characteristically downregulated at all the intervals in untreated RKN-inoculated plants, except at 0th DAI (Figure 4A).

### 3.5. PR1 Proteins

With respect to various treatments, the PR1 gene was characteristically downregulated in all the treatments on the 0th day, except in the healthy tomato rhizosphere. Upregulation of the transcript PR1 was noticed at the 3, 5, and 7th DAI in all the treatments, except in the inoculated control. The level of expression in the RKN-inoculated control was downregulated up to the 5th DAI of RKN, whereas a 0.2-fold upregulation of the transcript was noticed at 72 h. The maximum level of induction of the PR1 gene was observed up to 2.05-fold in the tomato rhizosphere soil drenched with *B. velezensis* VB7 challenged with *M. incognita*. It was subsequently followed by a 1.67-fold increase in *B. velezensis* VB7 alone applied soil at 7 DAI. Further, upregulation of PR1 was noticed in the untreated healthy control at all the intervals with a lower level of PR1 transcript than in the rest of the treatments. The plants treated with bioagents inoculated with RKN performed better in the induction of PR1 transcript rather than *B. velezensis* VB7 (Figure 4B).

### 3.6. Lipoxygenase (LOX)

Upregulation of LOX transcript was maximally induced in tomato plants applied with *B. velezensis* VB7 challenged with RKN compared to the inoculated and healthy control. Initially, the level of expression in the RKN-inoculated control was downregulated up until 3 DAI, whereas the level was upregulated on the 5th DAI and again downregulated on the 7th DAI. The plants treated with *B. velezensis* VB7 inoculated with RKN increased the LOX transcripts at all intervals after soil drenching with bioagents. LOX transcripts were induced up to 1.8-fold in the *B. velezensis* VB7 with the presence of RKN at 7 DAI. This was followed by 1.64-fold PR1 expression at 3 DAI after treatment. However, in the healthy control, transcript levels were increased up to 0.1-fold till 3 DAI, and subsequently, the transcripts were downregulated on 5 and 7 DAI (Figure 4C).

### 3.7. Myeloblastosis Related Proteins (MYB) TFs

MYB TFs is vital for the regulation of several physiological and biochemical processes in plant systems, including phenylpropanoid metabolism, plant hormone responses, and defense responses during pathogen interaction. Irrespective of the treatments, the expression of MYB transcript level was slowly upregulated from 0 DAI to 5 DAI. However, it was downregulated at 7 DAI which was also observed in healthy control. The transcript level was drastically raised by up to 2.2-fold in plants treated with *B. velezensis* VB7 inoculated RKN at 5 DAI; the level was not maintained as it was downregulated at 7 DAI. Further, soil drenched with *B. velezensis* VB7 had a 1.47-fold increase in MYB, whereas, in the absence of a challenge inoculation with RKN, it was decreased at 7 DAI. At all intervals, the MYB level was raised in the healthy control. However, in the RKN-infected control, the level of MYB increased initially (0.1-fold) at 0 DAI, and subsequently, it was downregulated at 3 and 5 DAI and slightly increased (0.3-fold) at 7 DAI (Figure 4D).

### 3.8. Phenylalanine Ammonia Lyase (PAL)

The transcript level of PAL was downregulated in the RKN-inoculated control for up to 72 h. At 0 h for all the treatments, PAL was downregulated, except in the RKN-inoculated control. The transcript level in *B. velezensis* VB7 + RKN, *B. velezensis* VB7 alone treatments was upregulated and was maintained up to 7 DAI. The transcript level of PAL increased from 1.10 to 2.25-fold in the rhizosphere soils of tomato plants treated with *B. velezensis* VB7 challenged against RKN. Similarly, even in the absence of RKN, the transcript level of PAL increased up to 1.9-fold in tomato plants drenched in *B. velezensis* VB7 alone at 7 DAI. However, the activity of PAL was downregulated at 5 DAI in the healthy control (Figure 4E).

### 3.9. Protein-Protein Interactions

The conserved domains of defense-related genes were clustered together in a network, which was analyzed using STRING to understand more about their functional partners. In the present study, the protein–protein interaction of WRKY revealed the existence of a broad protein family with a wide range of targets; its interaction with Brassinosteroid insensitive protein was associated with a phosphorylation-mediated signaling cascade of several novel transcription factors, including those of the protein kinase family, E3 Ubiquitin, protein kinase and other phosphatase enzymes involved in immunity. The cooperative and antagonistic functional interactions of WRKY factors were associated with the functions responsible for the enhancement of the defense response in plants (Figure 5A). During the interaction, the PR1 proteins were co-associated with LOX, PAL, jasmonate synthase, and MAMP Kinase and were linked to the regulation of immune responses against the infection of the pathogen (Figure 5B). Furthermore, LOX was interconnected with phospholipase, catalase isozyme, Putative ethylene protein, and with conserved domains in LOX linked with amino acid residues of lipid-associated proteins (Figure 5C). The MYB proteins also co-exist with SOD, Catalase isozyme, and other putative proteins (Figure 5D). PAL regulates the synthesis of many phenylpropanoid derivatives from phenylalanine and was responsible for resistance against nematode infection by producing lignin, and it served as a physical barrier against infection from RKN. Further, PAL was co-associated with AMP-dependent synthetase and ligase binding protein, 4 coumarate CoA ligase, and Trans cinnamate-4-monooxygenases (Figure 5E).

The functional enrichment analysis (FEA) provided insight into potential targets to improve the resistance of tomato plants to various biotic and abiotic stresses. The merging of WRKY33, PAL, PR1, MYB, and LOX resulted in the formation of an enlarged network comprising the interlinking of many nodes. Functional enrichment analysis resulted in the formation of more than five clusters. A heat cluster map was generated based on the functional relevance of each of the proteins. In the enrichment map, each cluster was linked with the co-expressed protein to indicate the commonality and the difference observed with the five defense proteins and their interaction partners. Based on the enrichment map, domain, and pathway information of the five proteins, LOX was linked with linoleic acid metabolism, oxidoreductase, dioxygenase, lipoxygenase, phospholipase, and PLAT/LH2 conserved domains. PAL was linked with the AMP binding enzymes, ubiquinone biosynthesis, CoA-ligase, chalcone synthase, polyketide synthase, and thiolase-like domains. The PR1 gene was associated with carboxylic acid biosynthesis, fatty acid metabolism, and oxidoreductase activity. MYB was coordinated with peroxidase activity and the fatty acid biosynthesis process. The WRKY transcription factor was associated with heterocyclic compound binding and the cellular anatomical entity, which are responsible for regulating the defense mechanisms (Figure 6).

## 4. Discussion

Root colonization by endophytic bacteria, also considered a Plant Growth Promoting Rhizobacteria (PGPR), is a safe alternative for the management of plant-parasitic nematodes [36,37,38,39,40]. Colonization of roots by bacterial endophytes on the root system resulted in the reduction of the nematode population by disrupting the cuticle and eggshell of the nematode. Thus, it prevented infection and the development of viable inoculum [18]. In the present study, the morphological character of *M. incognita* was circular to oval-shaped with a high dorsal arch coupled with smooth and wavy striae without any lateral lines onto the Perennial Cuticular Pattern (PCP). Similar to our results, the perineal cuticular pattern of *M. incognita* in tomato was characterized by a dorsal arch, coarseness, and smooth-to-wavy striae, without any lateral ridges with a whorled tail [41]. Our phenotypic description of RKN was also found to corroborate the findings of other groups [42,43]. The efficacy of bacterial endophytes against the hatchability of juveniles and their mortality was tested at different concentrations of culture filtrates of *B. velezensis* VB7. Our results indicated that an increased concentration of culture filtrate from endophytes increased nematicidal activity by preventing hatching capacity, and in addition, it also increased its mortality rate. In our study, *B. velezensis* VB7 enhanced the nematicidal effect with a 97.65% inhibition of egg hatching and an 87.66% reduction in the mortality of the juveniles at 96 h compared to other *Bacillus* spp. Consistent with our findings, Tian et al. [13] reported that within 12 h of exposure of *M. incognita* juveniles to the *B*. *velezensis*-25 broth, there was a 100% mortality rate, and egg hatching was prevented. Similarly, Jamal et al. [20] and Tadigiri et al. [44] found that *B. amyloliquefaciens* Ba-14.5 followed by *B. subtilis* Bs-13.9 (82.33%) were suppressed egg hatching by 86.0% and 82.33%, respectively. Further these bioagents also enhanced the mortality rate of *M*. *incognita* J2s in addition to promoting plant growth. The culture filtrate of *B. firmus* also induced the mortality of J2 juveniles and prevented the hatchability of *M. incognita* eggs [45]. The application of *B. velezensis* BZR 86 suppressed the infection of *M. incognita* in cucumber and tomato plants while promoting plant growth [10]. *Bacillus* sp. and *Pseudomonas* sp. isolates reduced the number of egg masses and the population of *M. incognita* in soil (Vetrivelkalai,2019) [46]. Apart from the direct nematicidal action, Nematode Associated Molecular Patterns (NAMPs) activate MAPKs and the JA/SA signaling cascade, leading to the activation of genes involved in MAMP-triggered immunity [47]. Recently, many investigations [13,48,49,50,51] have revealed that *B. velezensis* triggers the genes associated with SA and JA in plants and stimulates the defense response against nematode infection, resulting in reduced disease progression in various crops. Similar to these studies, the present study also found that the genes associated with the synthesis of SA and JA were induced by *B. velezensis* VB7 in tomato to defend against attacks from *M. incognita.* Likewise, the genes associated with defense responses were also activated in tomato plants co-inoculated with the bacterium and the nematode (Shukla et al. 2018).

In addition to the induction of defense genes, the transcription factor WRKY 33, coordinated with the expression of the defense response against pathogen infections [52], was also induced to enhance the immune response against RKN [53,54]. In response to the ingress of nematodes in the host plants, WRKY 33 also induced systemic resistance in the earliest phases of infection [55]. Similarly, we also observed the accumulation of WRKY33 transcripts associated with tomato defense against RKN. Hence, the applications of *B. velezensis* VB7 also activated a systemic resistance and thus have suppressed RKN due to the induction of transcription factors linked with defense genes contributing towards the suppression of the relationship between tomato and RKN. Similarly, the over-expression of WRKY 33 resulted in the induction of a resistance against *Heterodera schachtii* due to the activation of the MKK phosphorating cascade responsible for the synthesis of camalexin [56]. An increase in the expression of WRKY 33 transcript in tomato plants was due to the interaction of *B. amyloliquefaciens* against GBNV [32]. The up-regulation of WRKY11 and WRKY17 genes in *M. incognita*-infested *Arabidopsis thaliana* roots indicated that these transcription factors functioned as positive regulators for the activation of defense against RKN [57]. The WRKY23 was expressed during the early stages of feeding, leading to reduced infection by the cyst nematode *H. schachtii* [51]. There are several other reports indicating that the upregulation of WRKY genes triggered an immunity to biotic stress [54,58,59,60,61,62,63]. Similarly, the present investigation also confirmed the early induction of WRKY genes responsible for the suppression of RKN in tomato. The WRKY TFs induced NPR1, which in turn bound to pathogen-responsive cis-acting W-box promoter elements in PR1 genes responsible for the induction of PR1 and other defense-related genes (Kuźniak et al. [64] and Vanthana, et al. [65]. As PR1 played a significant role in the suppression of nematode infection in the host plants, the activation of PR1 in the tomatoes against RKN might have also suppressed the host nematode relationship in the tomatoes during the present investigation. Similarly, the PR transcript increased by 2.5 to 5-fold in tomato plants treated with *Bacillus* sp. and thus prevented the ingress of *M. incognita* [66]. The application of *B. velezensis v-25* to the cucumber plants induced the expression of PR1 and PR3 and thus suppressed the infection of *M. incognita* [13].

For the expression of the defense gene, PAL regulated the synthesis of phenylpropanoid derivatives from phenylalanine and served as a potential wound defender and physical barrier against nematode infections [67]. Antimicrobial phytoalexins formed by the interaction of PAL and other resistance-inducing co-associated proteins activated the defense enzymes that curb nematode infection. As per the earlier school of thought, we were also able to observe the multifold increase in PAL activity pertaining to the interaction of *B. velezensis* VB7 challenged with RKN. Thus, an increase in the activity of PAL transcripts might have not only increased the tomato plant defense, but also might have contributed to the suppression of RKN infection in addition to the direct nematicidal action of *B. velezensis* VB7.

MYB TF is vital for the regulation of several physiological and biochemical processes in plant systems, including phenylpropanoid metabolism, plant hormone responses or defense responses during pathogen interaction [68]. In response to pathogen infection, MYB activates hypersensitive cell death and thus increases plant defense through the Jasmonic acid-dependent pathway [69]. In the present study, transcript levels of MYB also increased during interaction with *B. velezensis* VB7. The infection of root-knot nematode in plants leads to rapid accumulations of JA in roots, and its subsequent transport to the leaves activates the plant’s defense mechanism against the nematode infection [70]. In plants, resistance against nematodes and other phytopathogens is also mediated through the induction of LOX transcripts. Plants that are deficient in the production of either JA or 12-oxo-phytodienoic acid (OPDA) are more vulnerable to nematode infection [71]. In our studies, the upregulation of the LOX gene was maintained up to 7 DAI, a higher transcript level was noticed in the plants treated with *B. velezensis* VB7 challenged against RKN. Thus, the increase in the activity of LOX transcripts might have promoted OPDA, leading to the synthesis of JA. The increase in JA might have resulted in the suppression of RKN apart from the direct nematicidal action imposed by *B. velezensis* VB7. Martinez-Medina et al. [72] and Ghahremani et al. [18] observed that the JA biosynthesis-related gene, Lox D, was up-regulated in tomato by the application of bioagents that inhibit the development of the nematode and its reproduction. *B. firmus* I-1582 and *B. amyloliquefaciens* QST713 stimulated the genes responsible for the induction of SA and JA in cotton to defend against *M. incognita* [73]. Similarly, the drenching of the tomato root zone with *B. velezensis* VB7 has also stimulated the gene transcripts responsible for the hormones associated with the ISR pathway.

The defense-related regulatory proteins in plants are co-induced in response to pathogen attacks, indicating the existence of a complex interaction with other proteins [74]. During nematode infection, the coordinated expression of several proteins was involved in the upregulation of the signaling of several phytohormones, including reactive oxygen species (ROS), MAPK-mediated cascades, and WRKY transcription factors responsible for the activation of constitutive and systemic defenses [59]. The protein–protein interaction studies clearly indicated the interaction of genes involved in defense responses with one another through their cross-linkages, which were visualized in the STRING database. The protein–protein interaction network can be constructed within the genes involved in resistance as well as transcription. Defense-related phytohormones such as salicylic acid (SA) and Jasmonic acid were triggered in response to pathogens and elicitors (JA). WRKY factors have a vital part in stimulating plant resistance to necro-trophic pathogens through the co-expression of defense genes via different biosynthetic pathways contributing towards the synthesis of antimicrobial phytohormones [75]. The tomato plants treated with *B. velezensis* VB7 challenged with RKN induced the expression of defense gene transcripts pertaining to PR1, PAL, LOX, WRKY 33, and MYB. These transcripts might have concurrently triggered several other pathways and proteins responsible for the inhibition of RKN infection via the induction of SA, JA, and several transcription factors complementing to overcome the biotic and biotic stress. Similar to our result, the plant resistance (R) proteins interact with WRKY TFs and R-WRKY proteins, indicating transcriptional regulation for fast immune responses [55]. Transcription factor MYB 30 encodes an activator for hypersensitive cell death in response to pathogen attack, acting through the regulation of very long-chain fatty acid synthesis and mediating the JA response [69]. MYB also induces the defense response by co-interaction with SOD and catalase enzymes in relation to nematode infection. The SA-responsive PR-1 and PR-5 genes were used as molecular markers for SAR activation in tomato plants [76]. PR1 proteins interacted with MAPK, LOX, and EDS 1 and formed a cross-linkage with conserved domains that suppressed the nematode infection due to the activation of the SA response. Supporting to our finding, the tomato plants inoculated with J2 of *M. incognita* enhanced the transcript level of PR1 and PR5 with respect to nematode attack [21]. The upregulated LOX gene associated with JA synthase and catalase enzymes in bioagent-treated plants might induce plant immunity against RKN [77]. Similar to our findings, the expression of PAL, MAPK, WRKY 33, CERK, and LOX transcripts were enhanced in banana plantlets bio-hardened with *B. velezensis* YEBBR6 against Foc KP, resulting in the simultaneous promotion of multiple pathways and proteins necessary for Fusarium wilt reduction in banana through the induction of both SAR and ISR [78]. Functional Enrichment Analysis (FEA) of defense genes was performed to identify the functional categories and biological pathways associated with defense genes in tomato plants. By identifying the pathways and processes involved in plant defense, FEA can provide insight into potential targets for improving the resistance of tomato plants to various biotic and abiotic stresses. In the present study, defense genes were coordinated with core proteins that triggered an immune response in tomato plants. Our study was in line with [78]. The differentially expressed genes related to tomato resistance to root-knot nematode and identified pathways were linked to plant defense response, including phenylpropanoid biosynthesis and plant hormone signal transduction pathways identified through functional gene analysis [79]. Consequently, the present study demonstrated that the application of *B. velezensis* VB7 inhibited the infection of RKN and induced MAMP-triggered immunity via transcription factor and resistance gene induction. The co-expression of the domains involved in innate immunity was further validated by protein–protein interactions and functional enrichment analysis, which confirmed that the application of biocontrol agents had the ability to induce a defense response that resulted in the suppression of RKN infection in the tomato plants. Therefore, the application of *B. velezensis* VB7 can enhance innate immunity, resulting in the suppression of infection by RKN in tomato plants.

## 5. Conclusions

This investigation was carried out to understand the efficacy of the potential bacterial endophyte *B. velezensis* VB7 on the hatchability of nematodes and its ability to induce an innate immune response in tomato plants against an infection of *M. incognita*. The culture filtrate of *B. velezensis* VB7 had nematicidal activity against *M. incognita* at different concentrations. The applications of *B. velezensis* VB7 induced defense gene expression and thereby suppressed the infection of RKN in tomato plants. Protein–protein interaction and functional enrichment analysis also emphasized the role of different genes associated with plant defense against RKN. The experimental results confirmed the ability of *B. velezensis* VB7 to trigger the defense gene transcripts, including PR1. WRKY 33, MYC TFs, LOX and PAL are responsible for the suppression of RKN in tomato plants, considering that the deferrable attribute of *B. velezensis* VB7 can be explored as a novel bacterium for the management of RKN in tomatoes in field conditions. The present experimental strategy of understanding the expression of five defense genes that included WKRY 33, PAL, MYB, LOX, and PR1 has been found to be varied and not constant during all the intervals with responses to the application of bioagents confronted with RKN. The future investigation of the mechanism of metabolites from bioagents and their response to the host towards RKN infection will provide insight into their potential agricultural applications.

## Figures and Tables

**Figure 1 genes-14-01335-f001:**
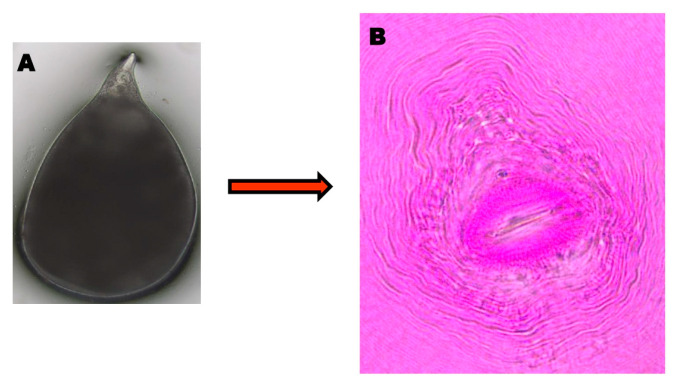
(**A**) Matured female of *M. incognita*, (**B**) posterior cuticular patterns (PCP) with typical characteristics of round to oval-shaped, high dorsal arch with substantial wavy striae and no lateral lines under the microscope.

**Figure 2 genes-14-01335-f002:**
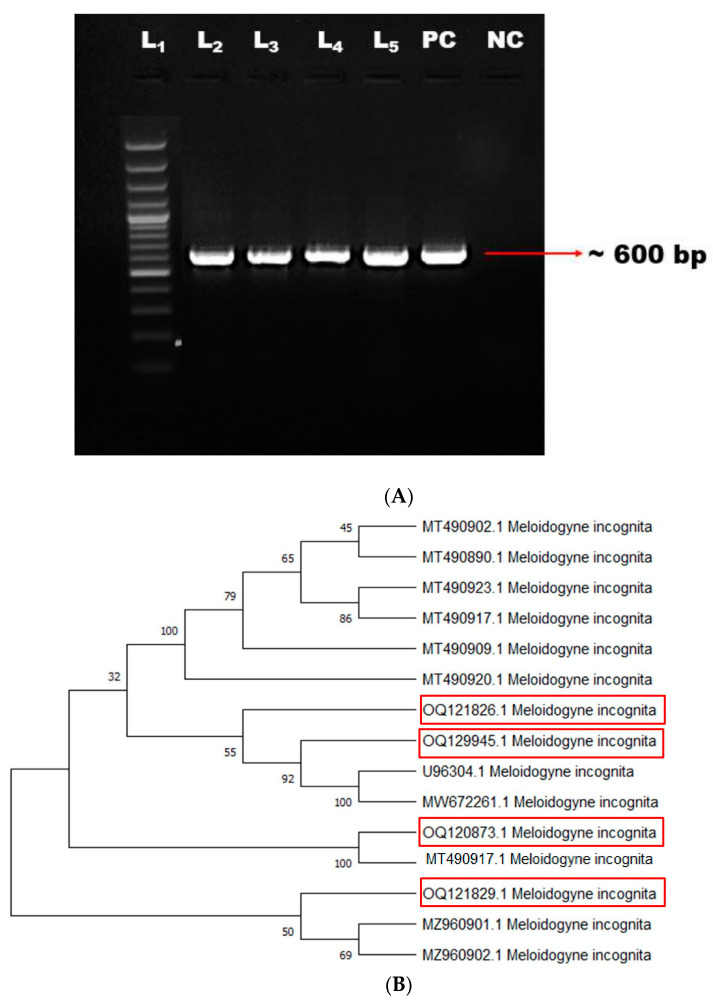
(**A**) Molecular characterization of *M. incognita.* L_1_—100 bp ladder, L_2_ to L_5_—*M. incognita* isolate with an amplicon size of ~600 bp PC—Positive Control, NC—Negative control. (**B**) Phylogenetic tree for *M. incognita.* The isolated *M. incognita* isolates were used in the present study are highlighted with red box.

**Figure 3 genes-14-01335-f003:**
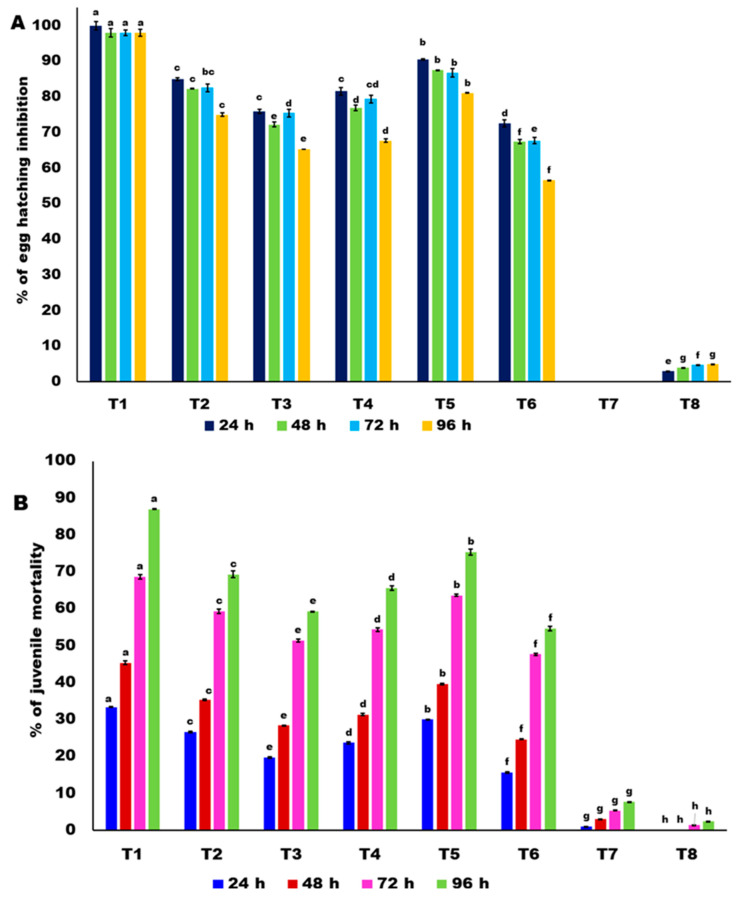
Efficacy of culture filtrate of bacterial endophytes on egg hatching and juveniles’ mortality of root knot nematode (RKN) *M. incognita* in vitro. (**A**) Percentage of inhibition of egg hatching. (**B**) Percentage of juvenile mortality. Error bars indicate the standard deviation obtained from three replicates. Analysis of variance was performed through DMRT. Means followed by different letters indicate significant differences (*p* < 0.05; *n* = 5) between treatments. T1-*B. velezensis* VB7, T2-*B. hyensii*, T3-*B. velezensis*, T4-*B. subtilis*, T5-*B. licheniformis*, T6-*B. subtilis*, T7-LB broth (Control), T8-Water (Control).

**Figure 4 genes-14-01335-f004:**
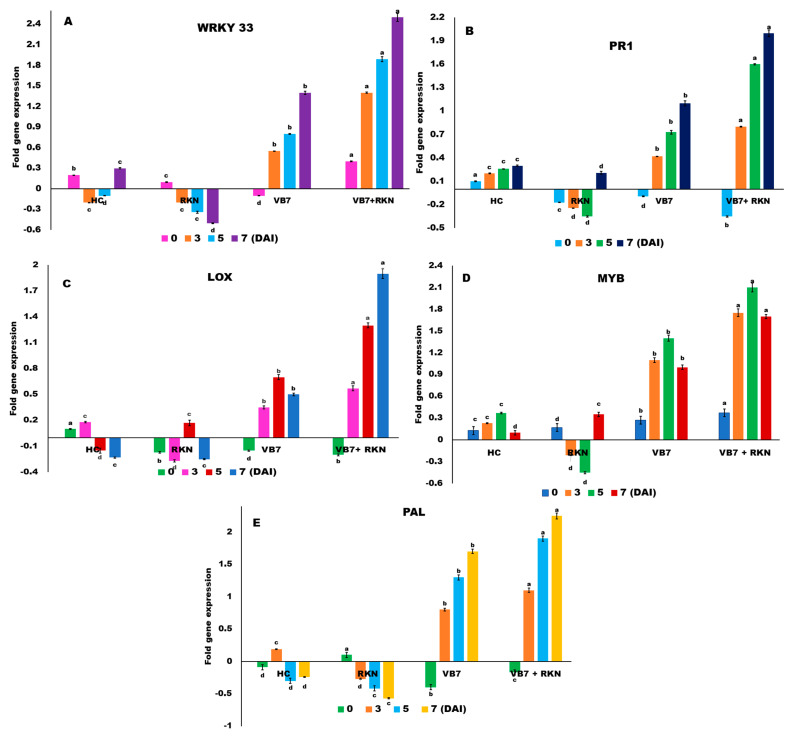
Expression pattern of defense genes and transcription factors. (**A**) WRKY 33 transcription factor, (**B**) Pathogenesis Related protein 1, (**C**) Lipoxygenase (LOX), (**D**) MYB Transcription Factor, and (**E**) Phenylalanine Ammonia Lyase defense genes in tomato plants treated with *B. velezensis* VB7; observations at different time intervals post-inoculation with RKN (0th, 3rd, 5th and 7th DAI) under mono-, di-, and tri-trophic interactions. HC—healthy control, RKN—inoculated control (Root Knot Nematode), VB7—*B. velezensis* VB7, VB7 + RKN—*B. velezensis* VB7 + RKN. Error bars represent the standard error of the mean values of three independent replicates. All treatments are significantly different from each other at *p* < 0.05. For each gene and time point, bars with different letters indicate statistically significant differences between treatments.

**Figure 5 genes-14-01335-f005:**
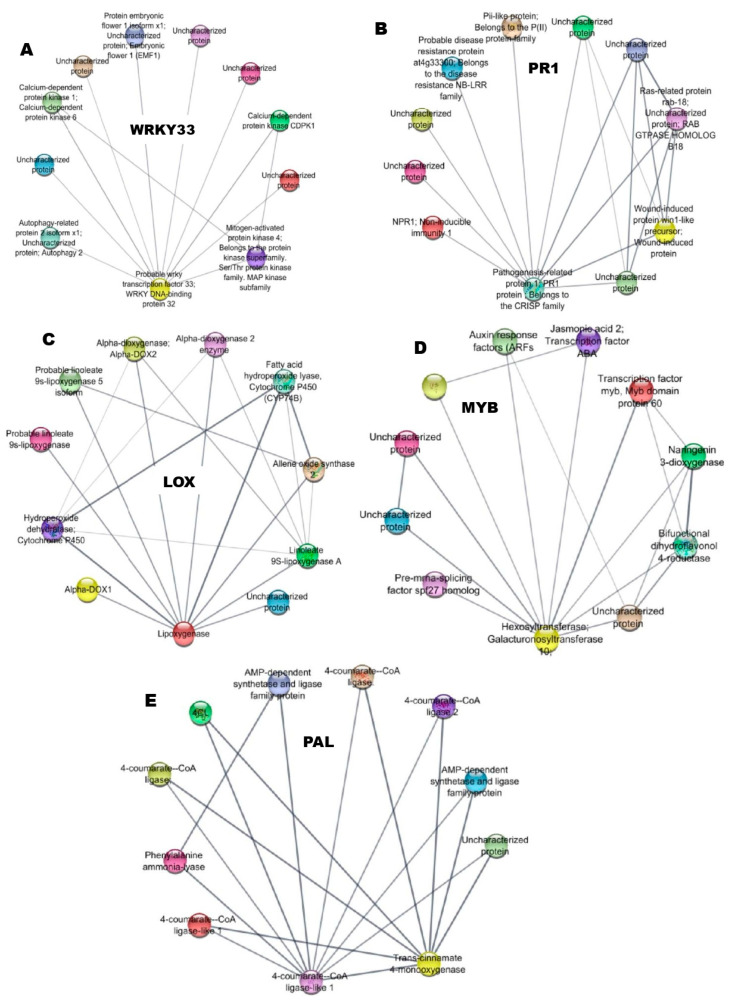
Protein–protein interaction analysis of defense genes (**A**) WRKY 33 transcription factor, (**B**) Pathogenesis Related protein 1, (**C**) Lipoxygenase (LOX), (**D**) MYB Transcription Factor, and (**E**) Phenylalanine Ammonia Lyase (PAL).

**Figure 6 genes-14-01335-f006:**
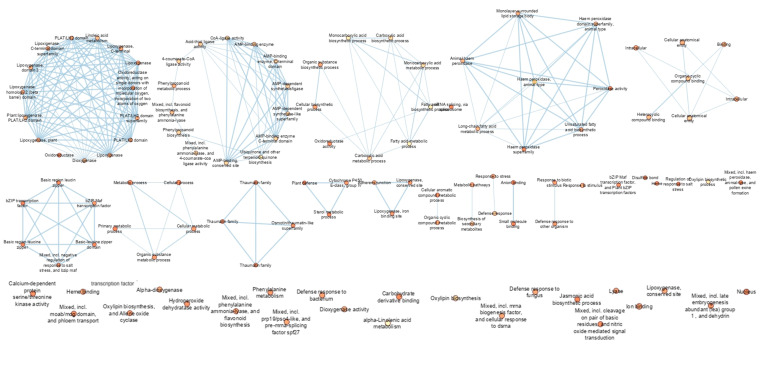
Functional enrichment analysis of defense genes during tri-tropic interaction in tomato plants.

## Data Availability

The original contributions presented in the study are included in the article/Appendix A, further inquiries can be directed to the corresponding author.

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
