# Peer review of "Antagonistic Bacteria Bacillus velezensis VB7 Possess Nematicidal Action and Induce an Immune Response to Suppress the Infection of Root-Knot Nematode (RKN) in Tomato"

_genes, 2023, doi:10.3390/genes14071335_

Round 1

Reviewer 1 Report

This paper is a study aimed at assessing the nematicide ability of a bacterial species of Bacillus. I find the study interesting and the findings with great potential of applicability in the control of root-knot nematodes. The experimental set-up, analysis, interpretation and presentation are sound in my opinion. There are a few things that authors should take care of.

1. I missed a comment on the real application potential of this data in nature since the experiments were done in vitro. How would you expect such an effect on the RKN to be under field conditions? Possibilities of prolonged immune response against RKN under field conditions.

2. You need to replace B. in the title with Bacillus.

3. Lines: 37-41; I found it hard to understand because past tense is used and got me confused.

4. Line 65-69: Sentence is not clear.

5. Line 72: Change to protein-protein.

6. Figure 1, 2, 5 and 6: Why are you repeating the legends two times? Please correct this error.

7. Figure 3: What do T1, T2... T8 mean? It is not defined!

8. Figure 4, 5 and 6: Quality is poor, consider re-doing in better version or format.

9. Line 373: PGPR is not the correct acronym to use in brackets there.

10. Would you consider impacts against hatchability a better strategy than direct impact on the RKN adults and why? 

Author Response

Thanks for the Reviewer's comment. As per your suggestion, we carried out the correction in the revised copy.

Reviewer 2 Report

This manuscript by Vinothini et al., attempted to analyse the effect of B. velezensis VB7 cell free supernatant on transcriptional changes in the immune defense of tomato plant during nematode infestation. Following are some queries and suggestions to authors.

1.     Bacterial species that has been chosen for this study is already demonstrated to act as a biocontrol agent against root knot nematodes. If so, what makes this study different and novel?

2.     On what basis the crucial defense genes were chosen for the real time analysis?

3.     How did authors choose to use B. velezensis VB7 for field applications? This part can be explained in the conclusion section.

4.     Authors should provide some data to validate the elicitation of immune response by cell free extract of B. velezensis VB7 in tomato plant, for example analyzing immune cells by flow cytometry.

5.     Did authors perform statistical analysis? P value or significance was not mentioned in any graph.

6.     Figure 6 is not readable.

7.     Drawbacks of the present study and future perspectives for the same should be detailed in the conclusion section.

8.     Did authors attempt to look at what biomolecule in the B. velezensis VB7 cell free extract shows the nematicidal activity?

9.     Authors should see the changes in phytohormonal signaling during nematode challenge in root knot disease and see if B. velezensis VB7 extract could alter this modulation.

Author Response

Thanks for the Reviewer's comment. As per your kind and valuable suggestion, the correction was carried out in the revised manuscript.

Reviewer 3 Report

Interesting article on the root nematode Meloidogyne incognita. These nematodes parasitize plants, causing damage to agriculture. These parasites alter root physiology and affect host phytohormonal signaling to evade defenses. The authors studied the nematocidal activity of B. velezensis to induce an innate immune response against nematode infection. As a result, hatchability of nematode eggs was found to be inhibited and therefore B. velezensis was shown to be effective as a potential nematocide and immune response inducer against RKN infestation in tomatoes.

Materials and methods described correctly.

Results clearly presented.

Figure 6 is not readable at all.

The references contain a substantial review of the literature.

Author Response

Thanks for the Reviewer's comment. As per your kind and valuable suggestion, we carried out the correction in the revised manuscript.

Round 2

Reviewer 2 Report

.